# Density or Connectivity: What Are the Main Causes of the Spatial Proliferation of COVID-19 in Korea?

**DOI:** 10.3390/ijerph18105084

**Published:** 2021-05-11

**Authors:** Yun Jo, Andy Hong, Hyungun Sung

**Affiliations:** 1Graduate School of Urban Studies, Hanyang University, Seoul 04763, Korea; whdbscjstk@naver.com; 2Department of City & Metropolitan Planning, College of Architecture + Planning, University of Utah, Salt Lake City, UT 84112, USA; a.hong@utah.edu; 3The George Institute for Global Health, Newtown, NSW 2042, Australia

**Keywords:** COVID-19, spatial proliferation, density, connectivity, social network analysis, negative binomial regression

## Abstract

COVID-19 has sparked a debate on the vulnerability of densely populated cities. Some studies argue that high-density urban centers are more vulnerable to infectious diseases due to a higher chance of infection in crowded urban environments. Other studies, however, argue that connectivity rather than population density plays a more significant role in the spread of COVID-19. While several studies have examined the role of urban density and connectivity in Europe and the U.S., few studies have been conducted in Asian countries. This study aims to investigate the role of urban spatial structure on COVID-19 by comparing different measures of urban density and connectivity during the first eight months of the outbreak in Korea. Two measures of density were derived from the Korean census, and four measures of connectivity were computed using social network analysis of the Origin-Destination data from the 2020 Korea Transport Database. We fitted both OLS and negative binomial models to the number of confirmed COVID-19 patients and its infection rates at the county level, collected individually from regional government websites in Korea. Results show that both density and connectivity play an important role in the proliferation of the COVID-19 outbreak in Korea. However, we found that the connectivity measure, particularly a measure of network centrality, was a better indicator of COVID-19 proliferation than the density measures. Our findings imply that policies that take into account different types of connectivity between cities might be necessary to contain the outbreak in the early phase.

## 1. Introduction

Since the outbreak of COVID-19 in December 2020, the virus has rampaged across the globe, causing 2.6 million deaths at the time of writing. South Korea has recorded a cumulated total of 92,817 positive cases and 1642 deaths. From its inception in China, the outbreak of the virus proliferated through Europe, the U.S., and eventually across all continents, at which point the World Health Organization officially declared it to be a pandemic [1]. The sheer speed of the virus infection and the consequent heavy death toll strike fear in all people around the globe. Based on previous research on infectious diseases, such as AIDS, SARS, and Swine flu (the 2009 H1N1 Pandemic), the key determinants of virus spread were found to be closely related to the movement of people, urbanization, and the mass inflow of foreign nationals from overseas, all of which are characteristics often found in large metropolitan cities [2,3,4].

While large cities offer many benefits, tightly connected networks of goods and people around urban cores could accelerate the COVID-19 pandemic. Matthew and McDonald [3] noted that the factors that precipitated the spread of malaria were economic development, patterns of land use, movement of people, and urbanization, whilst the spread of SARS and MERS were impacted by the inflow and outflow of people brought on by accelerated globalization. Other studies have also shown that transmission of infectious diseases is related to urban structure and people’s movement between cities. For example, SARS, a viral respiratory illness, first originated from a wild animal market in Guangdong Province but later spread via international air travel into the world’s cities. The 2014 West African Ebola virus shares similarities to SARS. Even though 64% and 60% of the population of Guinea and Sierra Leone lived in low-density agricultural areas, due to high connectivity and ease of travel, the two nations became the epicenter of the disease [5].

At first glance, places with a high degree of *urbanization*, such as New York, London, and Seoul, appear to be more susceptible to the proliferation of COVID-19. Previous studies have reported that infectious global diseases often leave behind a greater number of infections and a higher death toll [6]. Kao [7] found that the 2009 H1N1 virus lasted longer in areas with a higher population density within Taiwan. Garrett [8] also noted that there existed a statistically significant relationship between the mortality rate of the 1918 Spanish Flu (influenza) and the population density per state. Thus, the concentration of population in a given area seems to be a catalyst for spreading infectious diseases. On the other hand, Nishiura et al. [9] found that the 1918 Influenza in the UK and Japan showed no statistically significant relationship between population density, the severity of infection, and the mortality rate. Moreover, Parmet et al. [10] found that during the 1918 Influenza, areas of lower density saw higher mortality rates than that of high-density cities.

Given these mixed findings, this study aimed to investigate the role of urban spatial structure on COVID-19 during the first eight months of the outbreak in Korea. Specifically, we focused on two factors related to the built environment: density and connectivity. By comparing different measures of density and connectivity, we sought to provide empirical evidence to determine which of the two measures are more related to the spread of COVID-19 in Korea. This research helps inform public health agencies and municipalities to find more effective measures of prevention and response against COVID-19 and other infectious disease threats.

## 2. Literature Review

### 2.1. Cities and Pandemics

Globalization and urbanization have progressed rapidly over the past 100 years. Globalization, prompted by the development of transportation and communication technologies, has been recognized as a distinct shift in the spatial and temporal dimensions of social and economic life, a phenomenon known as “time-space compression” [11]. This transition to the highly connected global marketplace has not only facilitated the movement of capital but also people and information, overcoming both distance and temporal obstacles of physical movement. However, globalization has not only positive effects of opening up new opportunities and new markets but also negative effects of promoting imbalances between countries, regions, or social classes [2,4]. Moreover, globalization has led to an unexpected spread of disease by intensifying trade and human movement between countries [12,13], resulting in more frequent outbreaks of infectious diseases and their spread on a global scale. Before the rapid progress of globalization and urbanization, infectious diseases were more localized, and their spread was slow and intermittent [14].

Several high-profile infectious disease outbreaks occurred before the COVID-19 pandemic. For example, the 1918–1919 Influenza pandemic, known as the Spanish Flu, was the most severe pandemic in recent history, resulting in about 50 million deaths around the world. During this pandemic, the United States had implemented stringent social distancing measures, such as the closure of theaters, churches, schools, and saloons as well as the mandatory wearing of face masks [15]. Since then, with the exception of the 1968 Hong Kong influenza pandemic, the outbreak and spread of infectious diseases were limited to certain regions or countries and were contained before spreading around the world. However, the current COVID-19, although often compared to the 1918 Influenza pandemic, is not just the return of the typical pandemic [16]. This pandemic marks a new chapter in the history of infectious disease, demonstrating how quickly viruses can spread around the world in the 21st century with so many countries that are tied together in the global marketplace [16,17].

In both popular media and scholarly communities, there is an increasing debate around urban density and COVID-19 pandemic [18,19]. On the one hand, cities are often characterized by their high-density urban form, which could be a risk factor for infectious disease outbreaks, such as COVID-19 [20]. Additionally, urban activities are not confined within the city boundary or a certain spatial domain because cities are agglomeration centers for socioeconomic connections of people, goods, and services. Higher connectivity within a city or between cities can amplify the risk of infectious diseases. On the other hand, cities can also provide a critical line of defense against pandemics [21,22]. Responses to disease outbreaks could be faster and more efficient in larger cities with more resources and disaster preparedness. Cities have more diverse and higher quality health care systems, offering more choices and potential for their citizens to receive better care in times of crisis.

As we are responding to the immediate COVID-19 crisis, this pandemic highlights the need to start thinking about how to rebuild our cities and neighborhoods against future shocks. For example, urban planners have been increasingly discussing alternative urban models, such as “Superblocks”, “Tactical urbanism”, or “15-min City” to overcome some of the major problems that have been exposed during the pandemic [23,24,25,26]. Before we start thinking about the future of cities after the pandemic, we first need to come to the drawing board and make a good assessment of which characteristics of cities are more vulnerable to pandemics and how we might be able to address them.

### 2.2. Density and Connectivity

Many empirical studies have attempted to assess the effects of urban factors on the spread of infectious disease, but the results have been mixed. In a study of the 1918 Spanish Influenza (H1N1), Garrett [8] reported that there was a statistically significant relationship between mortality and population density and claimed that high-population-density counties served as a catalyst that accelerated the spread of the infectious disease. Arbel et al. [27] assessed the influence of population density and socioeconomic factors on COVID-19 infection rates in Israel and found that the possibility of COVID-19 infection grew with population density levels. Ehlert [28] attempted to find the relationship between density and socio-economic variables vis-a-vis mortality rate. He reported that population density and employment density were positively correlated with COVID-19 mortality rates. Wand and Li [29] found that population density mattered in the spreading of COVID-19 on the U.S. county level. They demonstrated that population density alone accounted for up to 76 percent of cumulative infection cases in the U.S. from early March to late May 2020. However, Agnoletti et al. [30], reported that density was not significantly associated with the distribution of COVID-19 in Italy. After reviewing the most up-to-date state of knowledge, UN-Habitat [22] reported that density in itself appears not to be a decisive factor in the spread of the virus; but rather, inequality around income, race, service provision, and pre-existing health conditions seemed to have played a critical role in aggravating the vulnerability of individuals.

Another important factor related to urban spatial structures concerns connectivity. In a study of the 1918 Spanish Influenza in England and Wales, Chowell [31] explored the association between influenza death rates and a number of geographical and demographical indicators. While they found that cities had 30–40% higher death rates than rural areas, there was little association between mortality and measures of population density or residential crowding. The World Health Organization [5] reported that even though the majority of the population of Guinea and Sierra Leone live in rural areas, high cross-border connectivity due to the ease of travel seemed to have caused the wide spread of Ebola, making the two nations the epicenter of the pandemic. In a recent study of the COVID-19 pandemic in the U.S., Hamidi et al. [32] found that urban density was unrelated to COVID-19 infection rates and even inversely related to death rates. They reported that connectivity between counties had a greater impact on both the infection and mortality rates than the population density of a given county. Similarly, reviewing research papers on the impact of urban density on COVID-19, Teller [33] argued that its impact was not closely related to density but connectivity.

### 2.3. Research Gap and Our Contribution

While there is a growing body of research examining the impact of urban spatial structure on COVID-19 outcomes, empirical evidence is mixed and inconclusive. These mixed results may stem from the use of different measurements across the study. Especially, the results could differ by how the dependent variable is measured and calculated. For example, in research conducted by Hamidi et al. [32], when the number of total confirmed COVID-19 cases was used as the dependent variable, the results for population density were found to be statistically significant. However, when the dependent variable was redefined as the number of confirmed cases per 10,000 population, the result was statistically insignificant. Therefore, it is important to assess how robust the results are to different definitions of the dependent variable.

Moreover, some measures of urban densities could be biased when the denominator includes undevelopable lands, such as mountain ranges and rivers. Hence, we should take caution while calculating population and employment densities by using administrative regions that only include inhabitable areas. The measure of connectivity could also affect study results. For example, Hamidi et al. [32] used a proxy measure of connectivity computed as a ratio of yearly airplane passengers per 10,000 population. While this is an appropriate measure in large countries with many domestic airline services, such measure may not be appropriate in smaller countries with limited airline travels. In addition, Toller [33] identified that only a few empirical studies measured connectivity to identify diffusion of COVID-19.

In assessing the impact of connectivity, the social network analysis has been used in the infectious diseases studies to identify geographic transmission hubs of the infection as well as geographic distribution of infected patients [34,35]. In addition, the analysis could be more useful for representing a holistic structure of the urban transportation network [36]. The indicators measured using their analysis method allowed for examining the degree of connection between nodes at the specified spatial unit which, in this case, are cities. Cho et al. [37] used the social network theory to analyze the characteristics of Chinese railway networks that have clear points of departure and arrival. The data on the points of departure and arrival for travel were used to compute mobility variation between cities. Shin [38] also analyzed the effects of COVID-19 on South East Asian airports’ network centrality and found that the density and centrality of the airport network have shrunk during the initial phase of the COVID-19 pandemic. We followed these social network methods to develop the indicators of degree centrality, closeness centrality, betweenness centrality, and eigenvector centrality.

In this study, we aimed to fill the gap in the literature by comparing different measures of urban density and connectivity during the first eight months of the outbreak in Korea. We make contributions to the existing literature on three perspectives. First, we used the actual density indicators that take into account only habitable lands. Second, we used the aggregated origination and destination (O/D) data for all travel modes to calculate the degree of connectivity, closeness centrality, betweenness centrality, and eigenvector centrality to gain insight into how the connectivity of the overall transportation network affected the spread of COVID-19. Finally, we tested the sensitivity of the results by using different measures of dependent variables. This study provides insights into multiple factors of urban spatial structure, reflecting the current debate at play on the vulnerability of densely populated cities.

## 3. Materials and Methods

### 3.1. Study Area

Our study area was the entire country of South Korea to empirically identify the main causes of the spatial proliferation of COVID-19. In response to the COVID-19 outbreak, the Korean government has adopted a stringent contact-tracing strategy, called the 3T framework (Testing-Tracing-Treatment), to control and reduce the number of COVID-19 cases [39]. Because of the effective control strategies and low infection rates, the Korean government has not imposed any national lockdown policies commonly adopted in the U.S. and most European countries [20]. Soon after the COVID-19 outbreak in Korea, the health officials swiftly installed drive- and walk-through testing facilities, adopted extensive movement- and contract-tracing strategies, and communicated movement trajectories of confirmed patients via online maps and automatic mobile phone alert systems [40,41]. Most Korean citizens actively followed the quarantine guidelines, such as mask-wearing and hand-washing, while voluntarily practicing strong social distance measures in high-incidence areas as a self-protection measure.

Moreover, private firms as well as public offices implemented flexible work schedules, such as working from home and flexible working hours, as well as encouraging their employees to hold virtual meetings in lieu of in-person meetings. A survey conducted by the Korea Enterprises Federation (KEF) [42] among the top 100 sales firms in September 2020 revealed that about 88 percent of the responded firms have implemented some form of work-from-home measures. Firms have adopted a wide range of work-from-home strategies to adapt to the rapidly escalating situation while continuing their operations remotely. Most of these behavioral and institutional changes were made possible relatively smoothly, largely owning to the past experience of the Middle East respiratory syndrome (MERS) outbreak, which caused great damage to the nation’s economy back in 2012. Korea is considered one of a few countries that has prevented the rapid spread of COVID-19 without implementing the national lockdown widely adopted by other high-income countries, while also sustaining the national health and economic systems [43].

Thanks to the Korean government’s rapid response and strict hygiene practices adopted by citizens and private firms, the infection rates have stayed well under control, with the rates ranging between 2 and 2.5% and even staying at 3.3% during the third spike in December 2020. This level is well below the 10% benchmark level that the World Health Organization recommends as adequate test-positivity levels [44]. In addition, a recent review of 29 COVID-19 studies reporting reproductive numbers [45] indicated that Korea was the only country that reported less than 1.0 in its estimated reproduction number (R, 0.76, 95% CI, 0.34–1.70). Other countries showed much higher reproductive numbers, reporting 3.14 for mainland China followed by 5.08 for Spain, 6.07 for Germany, and 6.32 for France.

The spatial unit of analysis in our study was a city, a county, and a district level, which is a smaller geographic unit than counties in the U.S. [32,46] and in Germany [28]. Several studies of COVID-19 have used similar geographic units (e.g., counties) to capture the effects of population density and connectivity on the spread of COVID-19. Moreover, the data on the number of confirmed cases for COVID-19 were only attainable at the city, county, and district levels, which are named Si, Gun, and Gu in Korean, respectively. While all of these administrative divisions have self-governing authority, they are classified differently depending on whether it is an urban or rural area as well as an urban area within a metropolitan area. Specifically, Si is an urbanized area, Gun is a rural area with small-sized towns, and Gu is an urbanized area within a metropolitan city in Korea. Figure 1 shows the spatial distribution of confirmed cases of COVID-19 at the county level as of 17 September 2020.

### 3.2. Main Outcomes

Our main outcome variable was the accrued number of positive cases starting from 20 January 2020 until 17 September 2020, collected per county, which is Si/Gun/Gu. There was no official county-level COVID-19 data, so we manually compiled the data from websites of each city, county, and district. The actual government websites from which we sourced the original data can be found in the Table A1 (accessed on 18 September 2020). The total number of confirmed cases by 17 September 2020 was 22,657. Of this number, 1432 cases were people returning from overseas, so we excluded them from the analysis. Moreover, the website for Seoul Metropolitan City, Busan Metropolitan City, Incheon Metropolitan City, and North Jeolla Province had a separate category for patients who contracted the disease from other regions, such as other cities or from foreign countries. We excluded these cases, making the total number of confirmed cases 20,787. 

### 3.3. Explanatory Variables

Of the variables that have the potential to affect the outcome variable, we chose indicators of density and connectivity as the explanatory variables. We calculated net density measures for the population and employment densities. Using the aggregated travel data with origination/destination(O/D) at the county level, social network theory was used to calculate the four different measures of connectivity. We further standardized the density and connectivity measures to enable robust comparison in the model results.

The urbanization-related indicators of density used in this research were population and employment densities. Data for these variables were drawn from the Korea Statistical Information Service (KOSIS). In order to accurately determine the effects of urban densities, we used the concept of net density, which refers to the number of housing units in a given area of land devoted to residential development. As discussed in the literature review, we took caution in calculating these density measures because the outskirts of urban areas in Korea are often dominated by forestry, agricultural lands, and other non-residential lands. Failure to account for these inhabitable lands could result in the underestimation of urban density [47].

We used the 2020 travel O/D data at the county level on all transport modes provided by the Korea Transport Database (KTDB) to compute the connectivity measures. We used social network analysis (SNA) to calculate these measures. The form of the analysis is usually either an array of the social networks or a structure classified by different social network characteristics [48]. A variety of social network analysis methods exist to quantify network structures, such as centrality, density, structural holes, concentration, and modularity.

In light of the COVID-19 pandemic, the analysis has been employed either to demonstrate the infection network of the confirmed patients and its structural characteristics [49,50,51] or to explore how key nodes can play an important role in social networks of COVID-19. These studies mainly relied on data from Twitter users and their followers [52] to understand the spatial diffusion of the COVID-19 infection [53] or to visualize the COVID-19 pandemic risk [54].

Our approach for the social network analysis is different from the aforementioned studies. We applied the social network theory to estimate the spatial structure and its functional characteristics, resulting from the movement of people in daily life using transportation O/D data aggregated at the county level. A city is a place where people travel within and move to and from, and it should not be confined by its administrative boundary [20,55]. Our goal was to identify the geospatial and relational characteristics which are embedded in the intricate and complex transportation networks and the movement of people influenced by their daily activities within the transportation networks [56,57].

We used the centrality measures to quantify how cities exist within a network [58]. Specifically, we calculated the indicators of degree centrality, closeness centrality, betweenness centrality, and eigenvector centrality. The equations to calculate each of the centrality measures are shown below in Table 1.

The degree of centrality is defined as the summation of the number of direct links upon a node, and, using the number of different links a singular node has, the degree of the node’s centrality can be quantified. In other words, degree centrality tells us if a node takes on a key role within the social network and is positioned in a place of importance. Cities with high-degree centrality can be interpreted as having high linkages, both direct and local, with other cities. Closeness centrality measures centrality based on the closeness of other nodes and differs from the degree of centrality in that it is calculated by adding the distances of the nodes that are both directly and indirectly linked. Since the distances between the nodes are considered, it is used to determine the global relationship of the social network as opposed to direct linkages. Betweenness centrality calculates where a node is situated in relation to other nodes, and if the value of centrality is high, the node can be understood as a mediating city between other cities. Finally, eigenvector centrality measures the comparative score of every node within a network based on the principle that connections to a node with high centrality must positively impact the centrality value of the node in question. In other words, a weight is placed on the connected node, which is then considered in calculating the node’s centrality. Cities with high eigenvector centrality have a high probability of being linked to cities with a high degree of connectivity, which in turn, implies that these cities are more likely to come into contact with other contagious viruses. It is also important to note that if a node is linked to a city with a low connectivity value, the eigenvector-centrality value can decrease.

### 3.4. Control Variables

Our control variables included physical environment, socioeconomic factors, and other additional indicators. Hamidi et al. [32] considered the vulnerable population, including the elderly, and the level of medical services and facilities, in addition to density, as the main factors affecting the spread of COVID-19 and mortality in the United States. Arbel et al. [27] and Ehlert [28] reported that socioeconomic factors other than density are closely related to the COVID-19 fatality rate. Identifying the effect of social distancing on trip reduction in U.S. metropolitan areas, Hamidi & Zandiatashbar [59] employed demographic attributes, park accessibility, as well as density. Teller [33], reviewing many empirical studies on the relationship between COVID-19 and urban density, reported that urban settlement environment, socio-economic factors, physical environment, and urban service level might be the main determinants of its spread. For example, access to parks can have more benefit than cost from the possible risk of COVID-19 disease outbreaks since they may play a role in mental health improvement [59,60] and adaption of people to not-crowed outdoor activities [61] during COVID-19. Agnoletti et al. [30] measured the unemployment rate as one of the control factors for the cases of COVID-19 by province.

We followed the methods used in previous studies to measure potential determinants of COVID-19 in Korea. Specifically, we included ratio variables for females and youth (20–29 years old) and the older adults (65+ years old) as demographic characteristics obtained from the KOSIS database. For neighborhood resources, measures of medical accessibility, nursing home service, and park accessibility were obtained from the public data portal (http://data.go.kr, accessed on 17 September 2020) and processed using ArcGIS version 10.6 (ESRI, San Diego, CA, USA), a geographic information system (GIS) software. Per capita growth regional domestic product (GRDP) was derived from the KOSIS database as a measure of neighborhood economic characteristics. Unlike the Italian study [30], we did not include the unemployment rate at the city-county-district level, as this measure was not available at the time of the study. Land area (km^2^) was also included to control for differences in the spatial units. In addition, there was a sudden rise in infection associated with a super-spreader event at a particular church in the city of Daegu. As shown in Figure 2, the confirmed cases from the Daegu metropolitan area accounted for the majority of accrued confirmed cases. Thus, we created a dummy variable to separately examine the effect of this particular region affected by the super-spreader event.

### 3.5. Analytical Approach

To estimate the numbers of COVID-19 cases over the study period, we developed a robust ordinary least squares (OLS) regression model and a robust negative binomial regression (NBR) model. The probability of being diagnosed with COVID-19 exhibits the same probability distribution of a coincidental event. In such a case, a Poisson regression model or a robust negative binomial model are more appropriate to use than a standard OLS model [62]. When using the Poisson regression model, the underlying assumption is that the mean and the variance of an event occurring are identical to each other. Hence, if the actual analysis of the data shows over-dispersion, then there are obvious limitations to using this model. As such, when looking at the means and the variances of the two dependent variables used in this study, we found that the Poisson model was not suitable. Over-dispersion was detected in the means of the total number of confirmed cases and cases per 10,000 (91.17 and 3.55, respectively) as well as their variances (28,435.98 and 66.51, respectively). Also, not many cases of zero-patients were observed in our data; therefore, it was more suitable to use a robust negative binomial model than a zero-inflated Poisson model [62]. Moreover, we chose to use the robust model to minimize the impact of heteroskedasticity.

Finally, this paper compared four models—two that differ by the type of model (OLS model and the NBR model) used and an additional two that differ in the definition used for the dependent variable (total number of positive cases and total number of positive cases per 10,000). Model A defined the dependent variable as the total number of confirmed positive cases, and the analysis was based on the OLS model. Model B, similar to Model A, used the same dependent variable but switched the analysis methodology to NBR. Model C, on the other hand, defined its dependent variable as the total number of positive cases per 10,000, and the OLS method was used for its analysis. Finally, we used the same dependent variable for Model D as we did for Model C but used the NBR method of analysis. All analyses were performed in Stata (version 16, StataCorp LLC, College Station, TX, USA).

## 4. Results

### 4.1. Descriptive Summary

Table 2 shows the summary statistics of the variables used in this research. The average number of confirmed COVID-19 at the city-county-district level was 91.17, and the confirmed cases per 10,000 population was 3.56. Their respective standard deviations were 196.05 and 8.16, which were more than twice the average. This indicates that there was larger between-group variability in the outcome measure. Also, the maximum values for cases and rate of COVID-19 were 1671 and 92.90, respectively, indicating non-normal distribution. The distribution of COVID-19 cases was skewed to the right (Figure 2); therefore, we standardized the density and connectivity indicators to allow direct comparison between the magnitudes of their impacts on the cases and their rates per 10,000 population for COVID-19.

### 4.2. Model Results

Table 3 shows the results of the OLS and NBR models. Two different measures of outcome were estimated: one for the total number of confirmed cases (Models 1 and 2) and the other for the number of confirmed cases per 10,000 people (Models 3 and 4). The indicator of dispersion, α was greater than zero, indicated that the data were over-dispersed. Hence, an NBR model appeared to be more suitable than a Poisson model.

In terms of the overall model fitness, the AIC and BIC values were smaller in the NBR models than the OLS models, indicating that the NBR models perform better than the OLS models. This implies that a random event could occur in the relationship, affecting the probability of infection. Hence, we focused our efforts on interpreting the NBR model results.

When it comes to the connectivity measures, degree centrality was most strongly correlated with the number of cases (coefficient = 45.62 for the OLS model and coefficient = 0.44 for the NBR model). However, none of the connectivity measures were statistically significant for the models that use infection rates. Of the density variables, only net population density was strongly associated with the number of cases (coefficient = 29.38 for the OLS model and coefficient = 0.31 for the NBR model). Again, none of the density measures were associated with the infection rates.

Among the population characteristics, the number of cases was positively correlated with the proportion of females (coefficient = 21.16) but negatively associated with the proportion of older adults (coefficient = −0.052) in the NBR model. The OLS model as well as the models that used infection rates showed no statistically significant results with any of the population characteristics. All of the variables indicating neighborhood resources were strongly correlated with the number of cases in the NBR model (Model 2).

While the availability of a nursing home was positively correlated with the number of cases, the number of doctors per 1000 people and park area per 1000 people were both negatively correlated with the number of cases, indicating some protective effects of neighborhood resources against the COVID-19 outbreak. This finding is similar to the conclusion of Ma et al. [61], who found that the benefits of visiting parks outweighed the risks associated with possible COVID-19 infection. Interestingly, the dummy variable indicating the city of Daegu was strongly correlated with both the number of cases and the infection rates across all models. This result indicates that the super-spreader event clustered around the Daegu area indeed was strongly associated with the spread of COVID-19 in the early phase of the outbreak.

## 5. Discussion

Our results suggest that the total number of confirmed cases was more appropriate than the infection rates in explaining the relationship with urban spatial structure. One possible reason is that using infection rates in the model could have resulted in canceling out the effect of population density. Because our models included density measures that incorporated population counts in the denominator, dividing the number of confirmed cases by population counts could negate the effect of density and other urban-related factors, leading to the classic type II error. This indicates that further caution will be needed to account for possible biases when modeling the spread of infectious diseases [63].

Using the standardized coefficients, we found that the measures of connectivity were more positively related to COVID-19 infection than the measures of density (standardized coefficient = 0.31 for net population density; standardized coefficient = 0.44 for degree centrality). This result is somewhat consistent with findings from previous research [32] that suggested that dense places do not necessarily lead to more infection but more connected places, as measured by a metropolitan size and an enplanement rate, were positively associated with the infection rates. Another study conducted in China also found that connectivity, as measured by the betweenness centrality of the road networks of 255 Chinese cities, was one of the key determinants of infection rates [64].

Interestingly, the eigenvector connectivity measure had a negative effect on the infection in our model. This was because the eigenvector centrality takes into account the level of connectivity of the connected node (city). Thus, even if the node is connected to a lower-networked city, it has a high degree of connectivity. Both closeness centrality and betweenness centrality were found to be statistically insignificant. This might be because the closeness-centrality measure captures the global connectivity of the social network; therefore, it might not fully capture all the variations in the network condition that are more related to the spread of the virus. Similarly, the betweenness centrality measures the degree to which a node is situated in between different cities within the social network. Again, we believe that this measure is irrelevant in explaining the network condition that is more vulnerable to virus dissemination. On the other hand, the degree of centrality which captures direct connection in the network was found to be the best predictor among other connectivity measures.

The net population density was positively and significantly associated with the virus infection. This is consistent with the results of previous studies that found a strong association between population density and infectious disease outbreak [8,27]. Interestingly, net employment density was not statistically significant in the model. This is somewhat different from the findings of Ehlert [28], where both population density and employment density were significant associated with the spread of COVID-19. The reason for this difference could be because Ehlert’s research was conducted in Germany and/or because he did not consider any indicators of connectivity in his research. In absence of the connectivity measure, employment density may actually capture the effect of the network connection between home and work. In addition, it is possible that our findings reflect the large-scale effect of social-distancing policies, where a transition to working from home gradually took place around August 2020. In fact, the Korean government never imposed strict lockdown policies that would have resulted in a complete closure of retail stores and businesses; however, our results suggest that activities and movements related to employment may have little impact on the spread of COVID-19. Further research should examine if there is a combined effect of social-distancing policies and employment density on COVID-19 outbreak.

### Limitations

There were a few limitations with the study. First, this study did not consider the influence of social-distancing policies implemented in Korea since February 2020. There were several versions of social-distancing policies throughout the study period, from school closure to restriction on social gatherings. We were unable to examine the effect of social-distancing policies because our data were based on a cumulative record of COVID-19 cases, which did not include time information. The individual public health agency by province only releases aggregate data; therefore, we compiled the data manually from individual government websites at one point in time (17 September 2020). Second, there is a slight possibility that the total number of cases in the early phase of the outbreak could be biased due to the availability of testing. However, some of the undercounting of confirmed cases were resolved as the government ramped up testing capacity early on. In fact, a number of international media praised Korea as one of the few countries with the most aggressive COVID-19 test program [65]. Third, this study did not consider the impact of COVID-19 mortality because no reliable data were available at the appropriate spatial scale. If high-quality mortality data become available, it would be desirable to include mortality rates along with infection rates in future studies.

## 6. Conclusions

This study provides evidence that connectivity seems to matter more than density itself. Contrary to the common belief, cities are not necessarily more vulnerable to the spread of the disease than the rural counterparts. Instead of focusing on restricting movements in all places, having stricter control over movements and activities in highly connected places could be more effective in dampening the spread of COVID-19.

We also found that direct and local degree centrality were more strongly associated with the spread of COVID-19 than the global closeness centrality. This suggests that future mitigation strategies may focus on restricting movements between cities that are directly linked. Despite having a lower effect size than the connectivity measure, the net population density was still statistically significant in explaining the dissemination of COVID-19. Therefore, we cannot ignore the fact that density, especially residential density, could still play a role in the spread of the disease.

Our study offers new insights into the current debate on the vulnerability of densely populated cities, especially from the perspective of an Asian country. If our results can be confirmed by prospective studies, they could have implications for developing mitigation strategies to contain future outbreaks. Connectivity and residential density may indeed reflect one of a constellation of factors that need to be taken into account for future planning to protect our cities against infectious disease threats in the foreseeable future. Our study provides important insights into how we might plan and build safer, healthier, and more resilient cities in a post pandemic world.

## Figures and Tables

**Figure 1 ijerph-18-05084-f001:**
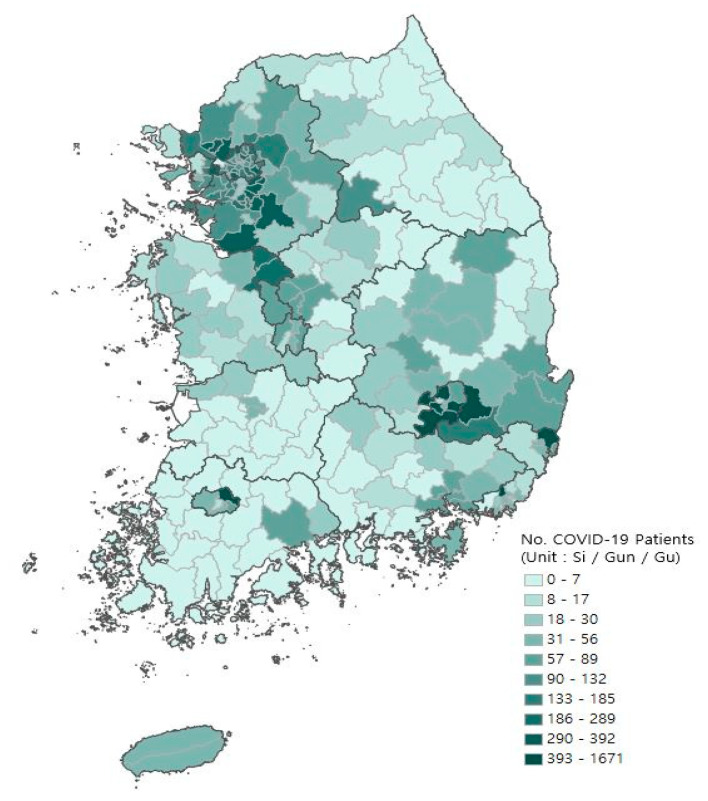
Spatial distribution of the number of confirmed COVID-19 patients at the county level (As of 17 September 2020).

**Figure 2 ijerph-18-05084-f002:**
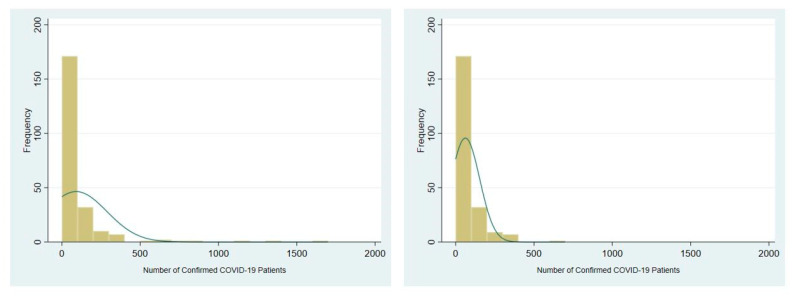
Frequency of COVID-19 cases with Daegu (**left**) and without Daegu (**right**).

**Table 1 ijerph-18-05084-t001:** The four indicators of connectivity.

Indicators	Equation
Degree Centrality	$\overset{g}{\underset{i=1}{\sum }}x_{ij},\;i\neq j$ $\sum_{i=1}^{g}x_{ij}$: Number of connections node *i* has with (*g* − 1) nodes.
Closeness Centrality	$\frac{1}{\sum_{i=1}^{g}d\left(N_{i},N_{j}\right)}$ $\sum_{i=1}^{g}d\left(N_{i},N_{j}\right)$: The sum of the closest distance between node *i* and *j* *g*: number of nodes
Betweenness Centrality	$\underset{i<k}{\sum }\frac{g_{jk}\;\left(N_{i}\right)}{g_{ik}}$ $g_{jk}$: the number of the closest routes between nodes *j* and *k* $g_{jk}\left(N_{i}\right)$: the number of closest routes between nodes *i* and *j* that includes node *i*
Eigenvector Centrality	$\lambda \overset{g}{\underset{i}{\sum }}x_{ij}C_{E}\left(N_{j}\right),\;i\neq j$ λ: Eigenvalue g: number of nodes x: the quantitative value of the connection between nodes *i* and *j*

**Table 2 ijerph-18-05084-t002:** Descriptive summary of the study variables (*n* = 228).

Variables	Description	Mean	Std. Dev.	Min	Max	VIF
*Dependent variables*						
Total cases	Accrued confirmed cases per city, county, district	91.171	196.051	0	1671	
Case rates	Accrued confirmed cases per 10,000	3.557	8.155	0	92.897	
*Connectivity*						
Degree centrality	Degree of centrality found using SNA	2.13 × 10^−9^	1	−1.101	4.599	4.59
Closeness centrality	Closeness Centrality found using SNA	−3.45 × 10^−9^	1	−5.77	1.361	1.85
Betweenness centrality	Betweenness Centrality found using SNA	−1.04 × 10^−9^	1	−0.514	10.257	1.39
Eigenvector	Eigenvector found using SNA	1.23 × 10^−9^	1	−0.182	10.356	1.46
*Density*						
Net population density	Population/Land Area	−1.33 × 10^−9^	1	−0.961	2.716	3.89
Net employment density	Total # Employed/Land Area	−1.58 × 10^−9^	1	−0.449	12.230	1.65
*Population characteristics*						
% Female	Total female population/total population	0.499	0.013	0.433	0.523	1.81
% 20 s years old	Total number of people between the ages of 20–29/total population	11.633	5.235	2.508	66.198	1.70
% 65+ years old	Total population of 65+/total population	21.556	8.277	8.7	40.7	4.78
*Neighborhood resources*						
No. of doctors/1000 people	(Total number of doctors/total population) × 1000	2.778	2.286	1	19.6	1.69
Availability of nursing home	Availability of nursing home (0/1)	0.552	0.498	0	1	1.18
Park area/1000 people	(Total park area/total population) × 1000	19,200.83	18,499.82	0	132,334.9	1.60
*Economic factor*						
Per capita GRDP	Growth regional domestic product (GRDP)/total population	33.677	30.935	8.072	385.763	1.57
*Other Control factors*						
Land area	Area of city, county, district (km^2^)	77.762	95.87	3.71	595.33	1.78
Dummy for Daegu	Whether or not include Daegu (0/1)	0.035	0.18	0	1	1.11

**Table 3 ijerph-18-05084-t003:** OLS and negative binomial regression (NBR) models of the district level COVID-19 cases (*n* = 228).

Variables	Number of Cases		Number of Cases/10,000 Residents	
	Model 1: OLS	Model 2: NBR	Model 3: OLS	Model 4: NBR
*Connectivity*				
Degree centrality	45.62188 * (−2.34)	0.44392 ** (3.18)	−0.50209 (−0.54)	0.15022 (−1.35)
Closeness centrality	−21.27929 * (−2.58)	−0.17435 (−1.40)	−0.44005 ^+^ (−1.86)	−0.12804 (−1.48)
Betweenness centrality	7.78549 (1.35)	0.30548 (1.45)	0.55370 (−0.75)	0.27804 (1.28)
Eigenvector	−9.26285 (−1.50)	−0.10502 ^+^ (−1.91)	0.22048 (0.68)	−0.04594 (−0.98)
*Density*				
Net population density	29.37699 * (2.51)	0.30569 * (−2.48)	−0.02309 (−0.05)	0.05392 (0.48)
Net employment density	−4.33808 (−0.70)	−0.00395 (−0.06)	−0.45149 (−1.04)	−0.01695 (−0.34)
*Population characteristics*				
% Female	562.86792 (−1.09)	21.16410 ** (2.60)	47.50386 (−1.41)	9.38009 (1.46)
% 20 s years old	2.66258 (−0.99)	0.00978 (−0.34)	0.22942 (−1.09)	0.01577 (0.89)
% 65+ years old	1.18664 (0.98)	−0.05200 * (−2.52)	−0.01805 (−0.29)	−0.00268 (−0.15)
*Neighborhood resources*				
No. of doctors/1000 people	−13.07725 (−1.49)	−0.05617 * (−2.12)	0.14679 (0.77)	−0.02717 (−1.16)
Availability of nursing home (0/1)	10.23637 (−0.8)	0.45998 ** (2.73)	−0.51969 (−0.62)	0.14658 (1.01)
Park area/1000 people	−0.00042 (−1.54)	−0.00001 *** (−3.69)	−0.00003 (−1.49)	−0.00001 (−1.58)
*Economic factor*				
Per capita GRDP	−0.19593 (−0.99)	−0.00176 (−1.21)	−0.002 (−0.36)	0.00067 (−0.56)
*Controls*				
Land area (km^2^)	−0.00914 (−0.11)	0.00113 (−1.43)	−0.00358 (−1.46)	−0.00139 * (−2.01)
Dummy for Daegu (0/1)	800.48779 *** (5.89)	2.45792 *** (−7.9)	32.33786 *** (4.63)	2.60876 *** (−10.8)
Constant	−228.62082 (−0.92)	−5.89396 (−1.51)	−22.74795 (−1.32)	−3.79063 (−1.24)
Log Likelihood	−1.37 × 10^3^	−1.05 × 10^3^	−694.8584	−465.41374
Adjusted R^2^ or Pseudo R^2^	0.73776	0.10397	0.57988	0.14277
Alpha (SE)		1.007541 *** (−0.1174)		0.493818 *** (−0.14)
AIC	2764.2	2141.95	1421.72	964.83
BIC	2819.07	2200.24	1476.59	1023.13

Note: *** *p* < 0.001, ** *p* < 0.01, * *p* < 0.05, ^+^
*p* < 0.10.

## Data Availability

The data underlying this research article are available in the Korean public data portal (https://www.data.go.kr/, accessed on 18 September 2020), Korean Statistical Information Service (https://kosis.kr/, access on 18 September 2020), and Korea Transport DataBase (https://www.ktdb.go.kr/, access on 18 September 2020). The COVID-19 data was available at the websites below (access on 18 September 2020).

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
