# Peer review of "Density or Connectivity: What Are the Main Causes of the Spatial Proliferation of COVID-19 in Korea?"

_ijerph, 2021, doi:10.3390/ijerph18105084_

Round 1

Reviewer 1 Report

The paper entitled “Density or Connectivity: What are the main causes of the spatial proliferation of COVID-19 in Korea” examines, through statistical analysis, the correlation of  COVID-19 infected cases in Korea with density  and connectivity, concluding that connectivity might be far more important than density in the COVID – 19 pandemic spread.

The subject of the paper is very interesting, and the research is well presented. The results of the research are clearly presented and could be very useful in order to define future planning strategies against a pandemic crisis. The manuscript is well written.

Please find below some comments for consideration:

I would suggest that the authors describe in short, the measures taken by the Korean state in order to reduce COVID-19 spreading. They have already included a remark on the research limitations section, but I believe it would be useful for foreign readers to know the framework of the lock down that was applied in Korea. For example, was there any measures for working from home or tele-working? Did this not affect for example the calculation of employment densities?

I would find it useful if the authors clarified the parameters that were considered for the calculation of the Indicators of Connectivity. Did they consider only railway connectivity, or did they take into account all possible transportation from each node to all others? Please clarify in line 138 the meaning “points of departure and arrival data”. Do you mean stations railway, bus, airports, highways?

Lines 387-388: The authors mention as a study limitation issue the possibility that the total number of cases in the early phase of the outbreak could be biased due to the availability of testing. I wonder if you considered also examining the death ratio in order to avoid this bias of testing that indeed exists in the calculation of infected cases.

Figure 2: the diagrams should be at the same scale in order to be comparable.

Line 188: TFor the purposes

Line 232: We fFollowing

Author Response

We appreciate for your valuable comments. Our authors have modified the manuscript based on your comments. The report on yours is attached as a MS word file because of the many contents. Please refer to the attached report for reflection on your comments.

Reviewer 2 Report

Overall, the paper shows good quality, English is very fluent but check again for very minor issues. The paper is worth publishing with just a major modification, i.e. the literature review.

The literature review must be strengthened: it is insufficient and very scarce in terms of references, i.e. this paragraph does not comprise a systematic (international) literature analysis. In this sense, I would suggest deepening your literature analysis with both historical and current perspectives. Take into consideration, for instance, the following ones:

(2020). Changing the urban design of cities for health: The superblock model. Environment International

(2020). Open issues and opportunities to guarantee the “right to the ‘healthy’ city” in the post-Covid-19 European city, Contesti

2020, ‘Post-pandemic’ transnational gentrifications: A critical outlook, Urban Studies

2021, A Methodological Approach on Disused Public Properties in the 15-Minute City Perspective, Sustainability

(2020). New Healthy Settlements Responding to Pandemic Outbreaks: Approaches from (and for) the Global City, The Plan Journal

(2020). COVID-19 and the City: It’s Smart to be Dense, https://www.itdp.org/event/covid-19-smart-to-be-dense/;

and create a literature review section divided into three parts. The first one should comprise the historical perspective on pandemics; the second part the trends in the current literature; and the third section the policies/proposal/actions of organizations such as OECD (Policy Responses to Coronavirus (COVID-19). Cities policy responses), C40 (https://www.c40.org/), etc.

The rest of the paper is very good, but to be published there is a step more to take (wrote the literature review).

Author Response

(The authors gave the same response as above.)

Reviewer 3 Report

Thank you for the opportunity to review your work.

2. Literature Review

It would be better to provide more details regarding connectivity. The definition might vary by research context. It would be beneficial to readers if this part includes: definition, methodologies, and how those methods are similar/different in the light of COVID19 study. 

3. Materials and Methods

Figure 1: as of "when"? Also, though the unit is similar to a county, it is not a county. it would be helpful to have the name of the unit on the legend/map.

Author Response

(The authors gave the same response as above.)

Round 2

Reviewer 2 Report

with this in-depth work, the paper is ready to be published. Congratulations!